# Deciphering the Roots of Pharmacists’ Critical Thinking About Pseudoscientific Claims: Insights from a Cross-Sectional Survey

**DOI:** 10.3390/pharmacy12060165

**Published:** 2024-11-06

**Authors:** Tomofumi Watanabe, Mari Matsumoto, Masami Ukawa, Makoto Ohira, Masaru Tsunoda

**Affiliations:** 1Division of Applied Pharmaceutical Education and Research, Hoshi University, 2-4-41 Ebara, Shinagawa-ku, Tokyo 142-8501, Japan; watanabe.tomofumi@hoshi.ac.jp; 2Faculty of Pharmacy, Iryo Sosei University, 5-5-1 Chuodai-iino, Iwaki, Fukushima 970-8551, Japan; matsumoto.mari@isu.ac.jp (M.M.); ukawa.masami@isu.ac.jp (M.U.); ohira.makoto@isu.ac.jp (M.O.); 3Graduate School of Life Science and Technology, Iryo Sosei University, 5-5-1 Chuodai-iino, Iwaki, Fukushima 970-8551, Japan

**Keywords:** pseudoscientific criticism scale, critical thinking, pharmacists, evidence-based medicine, continuing professional development, self-medication, public health

## Abstract

The global trend toward self-medication has increased public reliance on over-the-counter treatments and health-related information, contributing to the spread of pseudoscientific claims in healthcare and posing serious public health risks. Pharmacists, as accessible healthcare professionals, play a crucial role in critically evaluating these claims and providing evidence-based guidance. However, little quantitative research has assessed pharmacists’ critical thinking regarding pseudoscientific claims or the factors influencing them. This study aims to evaluate the demographic factors affecting pharmacists’ critical thinking about pseudoscientific claims. A cross-sectional survey was conducted among pharmacists in hospitals, insurance pharmacies, and drugstores across Japan. The newly developed Pseudoscience Criticism Scale (PCS) measured attitudes toward pseudoscientific claims. Statistical analysis identified factors that form and influence critical thinking. This study revealed two primary dimensions: “Medical Superstitions and Unscientific Treatments” and “Natural Healing Superstitions”. Gender and educational background significantly impacted PCS scores, with male pharmacists and graduates from six-year pharmacy programs exhibiting higher skepticism. These findings underscore the importance of ongoing professional development in pharmacy education to strengthen critical thinking. The PCS is an effective tool for assessing this competency. Enhancing educational efforts is essential to equip pharmacists to effectively counter pseudoscientific claims and improve public health.

## 1. Introduction

Pseudoscience comprises beliefs and practices that appear scientific but lack valid methodologies and supporting evidence [1,2]. Far from being harmless, pseudoscience often infiltrates social structures, exerting widespread influence [3]. In the public health field, pseudoscientific medical claims have become increasingly problematic [4,5,6,7], with dangerous interferences in medical practice leading to adverse outcomes, including deaths [8,9]. Moreover, the COVID-19 pandemic exacerbated this problem, and the widespread anxiety and fear associated with the pandemic encouraged the acceptance of evidence-poor information and treatments and contributed to the proliferation of pseudoscientific claims [10]. The impact of mass media and social media was particularly pronounced, and the unscientific hand sanitizers and infection control measures disseminated through these media contributed to a decline in public trust in public health.

The spread of pseudoscientific claims has also had a significant impact on self-medication practices [11,12]. The World Health Organization (WHO) defines self-medication as taking responsibility for one’s health and treating minor physical ailments by oneself [13], which includes not only self-management of minor illnesses using over-the-counter (OTC) medicines but also self-care for voluntary health management using health foods and supplements to maintain and promote good health [14]. Since the 1980s, self-medication has been encouraged in Western countries and more recently in Japan, driven by declining birthrates, an aging population, and rising lifestyle-related diseases [15,16,17].

In Japan, approximately 60% of the population uses health foods as part of self-medication, focusing primarily on product functionality, such as efficacy and effectiveness [18]. While health foods are intended to maintain and promote health, some imply disease prevention or treatment effects. This can lead consumers to mistakenly perceive these products as medications, using them in place of approved treatments or with unrealistic expectations of efficacy [19]. In the U.S., it has been reported that health supplements are used to improve health conditions even though they are clearly labeled as not intended to diagnose, treat, or prevent disease [8]. Such misinformed choices may directly endanger public health by delaying effective medical treatment and raising concerns about increased health risks [8,20]. Therefore, developing the skills to critically scrutinize pseudoscientific claims is a vital public health priority.

Pharmacists, as leaders in health and medical fields, are expected to assume expanded roles in self-medication consultation alongside traditional dispensing services and medication guidance [21]. Providing appropriate guidance and information is crucial for the safe practice of self-medication. Pharmacists must not only assist in selecting suitable OTC drugs and recommend medical consultations when necessary but also critically evaluate pseudoscientific claims and offer evidence-based information [21].

To the best of our knowledge, surveys have been conducted on pharmacists’ perceptions and attitudes toward pseudoscience [22], but these were mainly based on qualitative approaches, and no study has quantitatively evaluated critical thinking toward pseudoscience yet. Therefore, this study aims to clarify the structure of critical thinking that pharmacists employ when evaluating pseudoscientific claims. Specifically, we clarified the factors that lead pharmacists to form critical thinking toward pseudoscientific claims and developed a new scale (Pseudoscience Criticism Scale, PCS) to evaluate these factors quantitatively. Furthermore, we employed this scale to explore the underlying factors that influence the formation of critical thinking in pharmacists.

We hypothesize that critical thinking matures with age due to the accumulation of work experience and knowledge. Gender differences may influence health-related beliefs and behaviors, potentially affecting the development of critical thinking. Additionally, variations in work environments and professional roles may lead to differing exposures to pseudoscientific information and responses to it. The 2006 reform of the pharmacy education system in Japan, which extended the program from four to six years, likely enhanced educational content and may have promoted critical thinking skills among pharmacists. Based on these considerations, we identified four background factors for analysis: age, gender, educational program length, and industry of employment.

## 2. Methods

### 2.1. Study Design

Designed as a cross-sectional survey study, this study developed a new PCS based on data obtained from question items about pseudoscientific claims. To test the validity of the PCS, we first conducted exploratory factor analysis (EFA) to identify the factors constituting the scale, followed by confirmatory factor analysis (CFA) to evaluate its structural validity. Additionally, regression analysis was performed to assess the impact of these factors on the PCS quantitatively. The results provided a comprehensive assessment of the critical thinking structure regarding pharmacists’ pseudoscientific claims.

### 2.2. Research Instrument

The survey instrument used to assess pharmacists’ attitudes toward pseudoscientific claims was adapted from previous studies [12,23,24,25], with minor modifications. It comprised 16 pseudoscientific claims and 4 scientifically accurate statements (items 2, 5, 8, and 17). Participants rated each item on a 5-point Likert scale from 1 (disagree) to 5 (agree), with scientifically correct content questions treated as reverse items. The scores for each item were summed to calculate the PCS, which quantitatively measures the level of skepticism toward pseudoscientific claims. Higher PCS scores indicate stronger critical thinking and a more negative attitude toward pseudoscientific claims. The detailed structure of the questionnaire is shown in Table 1.

### 2.3. Participant Selection and Sample Calculation

Participants in this study were pharmacists working in hospitals, insurance pharmacies, and drugstores across Japan. A stratified random sampling method was employed based on health center jurisdictions to minimize regional bias, ensuring the homogenization of regional impacts and enhancing the generalizability of the findings. The required sample size was calculated to be approximately 400 participants, using a 95% confidence interval, a 5% margin of error, and an estimated population proportion of 50%.

### 2.4. Setting

On 28 April 2023, a request letter was mailed to the facilities where the participants worked, specifying the purpose and outline of the survey. Data collection was conducted using a web-based platform from 1 May to 30 June 2023. To prevent multiple responses from the same participant, each participant was assigned a unique personal ID, which they were required to enter when completing the survey. This unique identifier, along with checks for consistency in demographic data (age and gender), ensured that responses from each participant were recorded only once. Missing data were handled using the listwise deletion method.

### 2.5. Evaluation of the Pseudoscience Criticism Scale

The PCS scores were evaluated based on participants’ age, gender, pharmacy education program (four-year or six-year), and industry of employment. Comparisons between the two groups were performed using Student’s *t*-test. For comparisons among three groups, a one-way analysis of variance (ANOVA) was conducted, and when significant differences were observed, Tukey’s Honestly Significant Difference (HSD) post hoc test was applied. A *p*-value of less than 0.05 was considered statistically significant.

### 2.6. Psychometric Evaluation and Reliability Assessment

Item–total correlation analysis (I-T analysis) was conducted on the items related to pseudoscientific claims, and items with an I-T correlation of 0.3 or higher were selected [26]. EFA was performed using the maximum likelihood method and quartimin rotation, and the number of factors was determined based on scree plots. Only items with factor loadings of 0.35 or greater were retained, aiming for a simple structure. Items that showed factor loadings of 0.35 or greater for multiple factors were deleted [26]. The adequacy of the EFA was assessed using Bartlett’s test of sphericity and the Kaiser–Meyer–Olkin (KMO) measure of sampling adequacy (considered good if ≥0.6) [27].

To verify the internal consistency of the extracted factors, Cronbach’s alpha coefficients were calculated for the overall scale and each factor. A Cronbach’s alpha of 0.7 or higher was considered indicative of high reliability.

### 2.7. Confirmatory Factor Analysis

To validate the results of the EFA, CFA was conducted. The model fit criteria were set as follows: χ^2^/df (Chi-squared value/degrees of freedom) between 1 and 5, RMSEA (Root Mean Square Error of Approximation) ≤ 0.06, Tucker–Lewis Index (TLI) > 0.90, and Comparative Fit Index (CFI) > 0.90 [28].

### 2.8. Quantitative Evaluation by Regression Analysis

To quantitatively evaluate the impact of each identified factor on the PCS, a simple linear regression analysis (least squares method) was performed. Since multiple factors were involved, multiple regression analysis (least squares method) was also conducted to assess the extent to which each factor affected the PCS. The explanatory power of the regression model was evaluated using the coefficient of determination (R^2^).

### 2.9. Statistical Analysis

Item–total correlation analysis, EFA, demographic and factor score comparisons, and regression analyses were performed using JMP version 14 (SAS Institute Inc., Cary, NC, USA). CFA was conducted using R software version 4.0.3.

## 3. Results

### 3.1. Participant Characteristics

A total of 722 responses were obtained. After excluding 151 incomplete responses using the listwise deletion method, 571 valid responses were included in the analysis (valid response rate: 79.1%). Among the participants, 452 (79.2%) were between the ages of 30 and 59. Gender distribution included 335 males (58.7%) and 231 females (40.5%); 5 participants (0.8%) did not report their gender. Regarding educational background, 387 participants (67.8%) were four-year program graduates, and 184 (32.2%) were six-year program graduates. In terms of employment, 298 participants (52.2%) worked in hospitals, 238 (41.7%) worked in insurance pharmacies, and 35 (6.1%) worked in other settings or did not report their place of employment. A summary of the survey results is shown in Table 2.

### 3.2. Pseudoscience Criticism Scale Scores

The mean PCS score was 72.9 ± 10.7 (SD). Comparison by age (Figure 1A) showed that PCS scores tended to decrease with increasing age, but the difference was not statistically significant (*p* = 0.0580). Comparison by gender (Figure 1B) showed that males recorded significantly higher PCS scores than females (*** *p* < 0.001). In a comparison by pharmacy education program (Figure 1C), 6-year graduates had significantly higher PCS scores than 4-year graduates (*** *p* < 0.001). Comparison by place of employment (Figure 1D) showed no significant difference in PCS scores (*p* = 0.379).

### 3.3. Question Item Selection Through I-T Analysis

I-T analysis was conducted on 20 question items related to pseudoscientific claims. In total, 5 items (question items 2, 5, 8, 13, and 17) with I-T correlations less than 0.30 were excluded, resulting in 15 items selected for further analysis. Notably, 4 of the excluded items (excluding item 13) pertained to scientifically accurate content.

### 3.4. Extraction of Factors by EFA

EFA was performed twice on the 15 selected question items, and two factors were identified from the 13 question items. The factor loadings for all items met the criterion value of 0.35; the KMO index was 0.91, and Bartlett’s sphericity test was highly significant at *p* < 0.001, confirming that a highly reliable and valid instrument was constructed. The cumulative contribution of the identified factors reached 48.5%, exceeding the generally accepted standard of validity of 40%. The overall Cronbach’s alpha value was 0.85, Factor 1 was 0.81, and Factor 2 was 0.76, all high values, further confirming the reliability of the PCS. A summary of the EFA results is shown in Table 3.

### 3.5. Naming of Factors

Among the factors identified, Factor 1 included items related to superstitions and unscientific treatments related to medicine, such as “20. The antitumor effect can be expected by taking fucoidan as a supplement, which induces apoptosis of cancer cells.” and “14. The yin power of cabbage neutralizes the positive heat in the blood (thermal energy), so placing cabbage on your head can reduce fever.”, and was designated “Medical superstitions and unscientific treatments”. Factor 2 included items about misconceptions that natural forces and environmental factors directly benefit the body, such as “Soaking hands and feet in hot water infused with mineral germanium stimulates circulation, promotes recovery from fatigue, and improves a stiff neck.” and “Negative ions in the air promote our physical and mental health.”, and was designated “Natural healing superstitions”. The results of the naming of factors are shown in Table 3.

### 3.6. Confirmatory Factor Analysis

The two factors identified through the EFA were used as latent variables, and the 13 question items related to them as observed variables. The model demonstrated excellent fit indices: χ^2^/df = 2.71, RMSEA = 0.05, CFI = 0.95, and TLI = 0.94. The Akaike Information Criterion (AIC) was 20,627.62.

### 3.7. Regression Analysis

#### 3.7.1. Simple Regression Analysis

PCS was the dependent variable, and Factor 1 and Factor 2 were analyzed separately as explanatory variables. Factor 1 had a partial regression coefficient of 10.26 (R^2^ = 0.81), while Factor 2 had a partial regression coefficient of 10.16 (R^2^ = 0.70). This indicates that Factor 1 and Factor 2 are important predictors in the formation of the PCS and their influence in the formation of critical thinking regarding pseudoscience should be considered.

#### 3.7.2. Multiple Regression Analysis

The PCS was the dependent variable, and Factor 1 and Factor 2 were simultaneously analyzed as explanatory variables, with a standardized coefficient of 0.62 for Factor 1 and 0.43 for Factor 2. The results indicate that Factor 1 has a greater impact on the formation of the PCS than Factor 2. The variance inflation factor (VIF) was 1.74, indicating no multicollinearity issues [29]. The R^2^ for the overall model was 0.92, indicating that combining Factor 1 and Factor 2 significantly improved the explanatory power of the model. The results of these multiple regression analyses are presented in Table 4.

## 4. Discussion

This study is the first to quantitatively evaluate the newly developed PCS, designed to elucidate the structure of pharmacists’ critical thinking regarding pseudoscientific claims. The PCS serves as a valuable tool for assessing how pharmacists develop critical thinking skills and respond to pseudoscientific information, providing a quantitative measure of their attitudes toward such claims. The findings lay a foundation for enhancing pharmacists’ roles in public health and equipping them to effectively address pseudoscientific assertions.

Analysis of PCS scores revealed that pharmacists’ backgrounds significantly influence the formation of critical thinking. Specifically, gender and the type of pharmacy education program completed had substantial impacts. Male pharmacists demonstrated more critical attitudes toward pseudoscientific claims compared to female pharmacists. This result is consistent with a previous study of the general public by Aarnio and Lindeman [30], Pennycook et al. [31], and Majima [25]. In addition, Lindeman and Aarnio [32] and Pacini and Epstein [33] highlighted differences in analytical thinking styles by gender, and future studies should examine the cultural factors responsible for these differences.

Contrary to expectations, older pharmacists were more likely to accept pseudoscientific claims, suggesting that increased clinical experience does not necessarily enhance critical judgment and may contribute to confirmation bias or stereotypes. This finding indicates that continuous education is crucial, regardless of years of experience, to prevent complacency and ensure adherence to evidence-based practices. Pharmacists who completed the six-year pharmacy education program introduced in 2006 exhibited more critical attitudes than those with traditional four-year degrees. This suggests that the modernization of educational content may improve the quality of pharmacists.

To identify the underlying factors influencing critical thinking as measured by the PCS, EFA was conducted. Initially, I-T analysis led to the exclusion of five items from the twenty pseudoscientific claim questions, primarily those assessing scientifically accurate content. This outcome suggests that while pharmacists possess sufficient scientific knowledge, this knowledge alone does not significantly influence the formation of critical thinking toward pseudoscience. Instead, the ability to discern and critically evaluate pseudoscientific claims is more pivotal.

The EFA resulted in the selection of 13 items, from which two main factors were identified: “Medical superstitions and unscientific treatments (Factor 1)” and “Natural healing superstitions (Factor 2)”. Factor 1 represents the ability of pharmacists to critically address medical superstitions and unscientific treatments encountered in clinical practice. The prominence of this factor highlights the necessity for pharmacists to have accurate medical and pharmaceutical knowledge to debunk myths and guide patients appropriately. On the other hand, Factor 2 reflects beliefs in the inherent healing properties of natural elements or environmental factors without scientific validation. The presence of Factor 2 indicates that some pharmacists may hold unsubstantiated beliefs in alternative therapies, which could affect their professional judgments. This finding is consistent with the result of a previous study of Spanish pharmacists by Salvador-Mata and Cortiñas-Rovira22, suggesting that the issue of how pharmacists respond to pseudoscientific claims is a global challenge. These factors are instrumental in predicting pharmacists’ critical responses to pseudoscientific claims. The multiple regression analysis further revealed that Factor 1 has a more substantial impact than Factor 2 on the formation of critical thinking about pseudoscience. This finding emphasizes the critical role of confronting medical superstitions directly related to patient care.

The results of this study indicate that continually reinforcing critical thinking among pharmacists is essential to minimize the impact of pseudoscientific claims and maintain the quality of public health. It is particularly important to provide pharmacists with practical learning opportunities in continuing professional development (CPD) education programs, utilizing case studies and simulations to address pseudoscientific claims. This will enable pharmacists to go beyond mere knowledge transfer and further enhance their ability to think critically and analyze. Furthermore, the factors identified in this study will contribute to improving pharmacists’ ability to practice evidence-based medicine and further enhance their critical thinking skills. Educational programs on pseudoscience should be continuously developed and improved in response to changing times, and their effectiveness should be regularly evaluated. This process is expected to optimize the educational content and contribute to improving the quality of information provided by pharmacists.

This study opens several potential avenues for future research in understanding and strengthening pharmacists’ critical thinking regarding pseudoscientific claims. First, longitudinal studies could examine how pharmacists’ critical thinking skills evolve with sustained exposure to CPD programs incorporating PCS-based assessments, providing insights into the long-term impacts of educational interventions. Additionally, investigating the specific elements within CPD that most effectively enhance critical thinking would offer valuable direction for curriculum development. Future studies could also explore differences in critical thinking across various healthcare professions, comparing pharmacists with other professionals such as nurses or physicians to understand profession-specific needs in combating pseudoscientific information.

Several limitations exist in this study. First, the reliance on self-reported survey data may introduce response bias, as participants’ subjective perceptions could influence the results. Social desirability bias may have led some respondents to answer in a manner they perceived as professionally favorable. Second, since this cross-sectional study was conducted exclusively among pharmacists in Japan, the generalizability of the findings to other countries with different cultural, educational, and healthcare systems may be limited. Cultural attitudes toward pseudoscience and differences in pharmacy education curricula could affect the applicability of the results elsewhere. In addition, cross-sectional research approaches are limited to data from a specific point in time and have not been able to track changes in pharmacists’ critical thinking and attitudes over time. Future research should longitudinally examine the pedagogical effects of educational programs for developing critical thinking in CPD.

## 5. Conclusions

The PCS developed in this study was a useful tool for measuring pharmacists’ attitudes toward pseudoscientific claims and elements of critical thinking. Factor analysis identified two factors, medical superstitions and unscientific treatments and natural healing superstitions, as important components of critical thinking. This finding indicates that introducing continuous CPD in pharmacy education is essential to strengthen evidence-based critical thinking. This will enable pharmacists to minimize the impact of pseudoscientific claims and contribute to maintaining and improving the quality of public health.

## Figures and Tables

**Figure 1 pharmacy-12-00165-f001:**
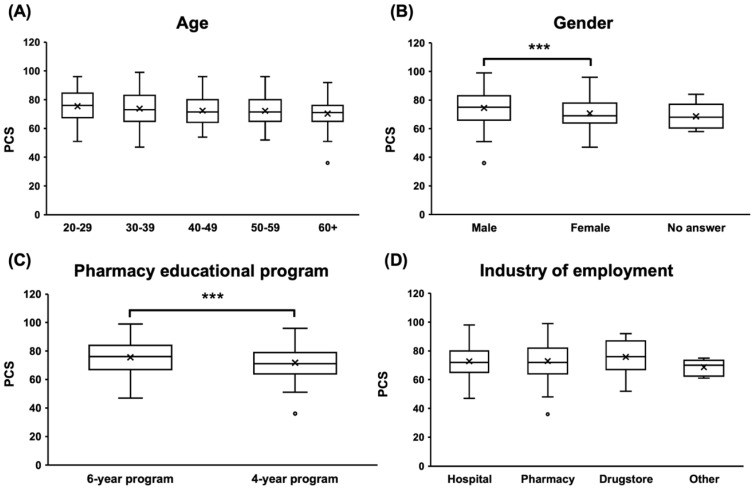
Group comparison of PCS scores. (**A**) Comparison by age showed a trend toward lower PCS scores with increasing age, but not significantly (*p* = 0.0580). (**B**) Comparison by gender showed significantly higher PCS scores for males than females (*** *p* < 0.001). (**C**) Comparison of 6-year and 4-year graduates by place of employment showed that 6-year graduates had significantly higher PCS scores than 4-year graduates (*** *p* < 0.001). (**D**) Comparing by place of employment, there was no statistically significant difference in PCS scores (*p* = 0.3787).

**Table 1 pharmacy-12-00165-t001:** Questionnaire items assessing pharmacists’ attitudes toward pseudoscientific claims.

Question Item
Listening to classical music, such as Mozart, makes us smart. Reactive oxygen species are generated within the body. Eating foods rich in collagen can improve your skin. Before an earthquake, sometimes irregular events occur, such as the water level increasing or decreasing, abnormal animal behavior, extraordinary weather, or malfunctions in communication devices. Tap water is safer than alkaline ionized water. Homoeopathic remedies foster spontaneous healing. Mobile phones use electromagnetic radiation in the microwave range; hence, mobile phone usage carries a risk of brain tumors. Even healthy people have many bacteria in their intestines. Excessive video game playing can cause damage to the prefrontal cortex. Eating hydrogen gum can reduce reactive oxygen species in the intestines due to the hydrogen ions it contains, balance intestinal bacteria, and improve bowel movements. An ionic detox bath that has an electric current passed through it draws out toxins from our body. The thinner the blood, the healthier it is. It is possible to control others’ behavior with subliminal messages. The yin power of cabbage neutralizes the positive heat in the blood (thermal energy), so placing cabbage on your head can reduce fever. Negative ions in the air promote our physical and mental health. Soaking hands and feet in hot water infused with mineral germanium stimulates circulation, promotes recovery from fatigue, and improves a stiff neck. Cholesterol is a substance necessary for building and maintaining human body functions. Nigari (magnesium chloride) has a beautifying and slimming effect. Exposure to natural radiation from granite can extend telomere length and potentially increase lifespan. The antitumor effect can be expected by taking fucoidan as a supplement, which induces apoptosis of cancer cells.

**Table 2 pharmacy-12-00165-t002:** Background information on survey participants (*n* = 571).

	*n* (%)
Age (in years)	
20–29	69 (12.1)
30–39	167 (29.2)
40–49	168 (29.4)
50–59	116 (20.3)
60+	51 (8.9)
Gender	
Male	335 (58.7)
Female	231 (40.4)
No answer	5 (0.9)
Pharmacy education program	
6-year program	184 (32.2)
4-year program	387 (67.8)
Industry of employment	
Hospital	298 (52.2)
Pharmacy	238 (41.7)
Drugstore	29 (5.1)
Other	6 (1.1)

**Table 3 pharmacy-12-00165-t003:** EFA results for pseudoscientific claims.

Item	Questions Regarding Pseudoscientific Claims	Factor Loadings		Communality
		Factor 1	Factor 2	
Factor 1: Medical superstitions and unscientific treatments				
19	Exposure to natural radiation from granite can extend telomere length and potentially increase lifespan.	0.81	−0.12	0.56
9	Excessive video game playing can cause damage to the prefrontal cortex.	0.72	0.04	0.55
20	The antitumor effect can be expected by taking fucoidan as a supplement, which induces apoptosis of cancer cells.	0.71	−0.02	0.49
14	The yin power of cabbage neutralizes the yang power of fever (blood), so putting cabbage on your head will reduce fever.	0.60	−0.03	0.35
11	An ionic detox bath that has an electric current passed through it draws out toxins from our body.	0.60	0.17	0.49
12	The thinner the blood, the healthier it is.	0.40	0.05	0.19
7	Mobile phones use electromagnetic radiation in the microwave range; hence, mobile phone usage carries a risk of brain tumors.	0.36	0.16	0.22
Factor 2: Natural healing superstitions				
4	Before an earthquake, sometimes irregular events occur, such as the water level increasing or decreasing, abnormal animal behavior, extraordinary weather, or malfunctions in communication devices.	−0.18	0.63	0.31
16	Soaking hands and feet in hot water infused with mineral germanium stimulates circulation, promotes recovery from fatigue, and improves a stiff neck.	0.28	0.58	0.58
6	Homeopathic remedies foster spontaneous healing.	0.23	0.50	0.42
15	Negative ions in the air promote our physical and mental health.	0.31	0.48	0.49
10	Eating hydrogen gum can reduce reactive oxygen species in the intestines due to the hydrogen ions it contains, balance intestinal bacteria, and improve bowel movements.	0.10	0.42	0.23
1	Listening to classical music, such as Mozart, makes us smart.	0.21	0.36	0.25
Eigenvalues		4.94	1.37	
Variance explained (%)		37.98	10.53	
Reliability: Cronbach’s α		0.81	0.76	
Overall Cronbach’s α		0.85		
The EFA identified two factors from the 13 items. Factor 1 was named “Medical superstitions and unscientific treatments” and Factor 2 as “Natural healing superstitions”. The factor loadings showed a criterion value of 0.35 or higher, with a KMO index of 0.91 and Bartlett’s sphericity test of *p* < 0.001. The cumulative contribution ratio was 48.5%, with an overall Cronbach’s alpha of 0.85. This confirms the reliability and validity of the PCS. The factor extraction method used was the maximum likelihood method, and the rotation method used was the quartimin method.				

**Table 4 pharmacy-12-00165-t004:** Multiple Regression Analysis Results.

Explanatory Variable	Unstandardized Coefficients		Standardized Coefficients	t	*p*	Statistics of Collinearity
	β	Std. Error	B			VIF
Factor 1	7.04	0.18	0.62	39.06	<0.0001	1.74
Factor 2	5.27	0.19	0.43	27.45	<0.0001	1.74
R^2^ = 0.92, Adjusted R^2^ = 0.92						
The results of the analysis of the influence of Factor 1 and Factor 2 on the PCS are shown. The standardized coefficients were 0.62 for Factor 1 and 0.43 for Factor 2, indicating that Factor 1 has a greater impact on the formation of the PCS (*p* < 0.0001). The VIF was 1.74 and indicated no multicollinearity issues. The R2 of the model was 0.92.						

## Data Availability

The data disclosed in this study are obtainable upon request from the corresponding author.

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
