# Peer review of "Deciphering the Roots of Pharmacists’ Critical Thinking About Pseudoscientific Claims: Insights from a Cross-Sectional Survey"

_pharmacy, 2024, doi:10.3390/pharmacy12060165_

Round 1

Reviewer 1 Report

Comments and Suggestions for Authors

Firstly, I would like to thank you for inviting me to participate in the review process of this manuscript, which represents an interesting, relevant, and well-conducted study.

I will provide a few comments on the article, focusing on different sections, with minor recommendations, as I did not identify any significant methodological issues that could compromise the robustness or reliability of the findings presented.

Introduction

The research topic and problem are well-framed and supported by the literature. The structure is logical and coherent, effectively leading to the formulation of the study’s hypotheses.

Materials and Methods

The methods selected are appropriate for the study's objectives and align with the central research question, including the methodological procedures related to the validation of the scale used.

I do have one clarification request. In the ‘Setting’ subsection, it is mentioned that ‘data collection was conducted using a web-based platform from May 1 to June 30, 2022, and was set up to prevent multiple responses by the same participant’. I would like to understand how this was managed—specifically, whether participants were asked to provide unique identifiers (e.g., email, registration number, personal ID, IP address) to ensure that multiple responses from the same participant were prevented. This clarification is important to address concerns around anonymity and compliance with data protection regulations.

Results

The results are presented in a coherent and clear manner, making it easy for the reader to understand and interpret the data.

Discussion

I would suggest adding a paragraph on the potential avenues this study opens for future research on the topic. In particular, it would be valuable to explore what new questions arise from this research and what directions could be pursued in subsequent studies.

Conclusion

I believe there is a need for a separate concluding section that synthesises the main findings of the manuscript, providing a summary of the study's contributions.

Author Response

添付ファイルをご覧ください。

Reviewer 2 Report

Comments and Suggestions for Authors

Overall, the document reflects the good work of this study, but there are nuances that should be improved in the document:

Lines 109-113 should be deleted (this paragraph has been written before).

The first word of paragraph 258 (This) should be spelt correctly.

Be consistent and write I-T analysis, instead of item-total correlation (line 286), to be consistent with the acronym used above.

Avoid repetition: ‘Medical superstitions and unscientific treatments (Factor 1)’ and ‘Natural healing superstitions (Factor 2)’. Factor 1 represents the ability of pharmacists to critically address medical superstitions and unscientific treatments encountered in clinical practice.’ Lines 293 to 295

Author Response

添付ファイルをご覧ください。
